# Preclinical Study in Mouse Thymus and Thymocytes: Effects of Treatment with a Combination of Sodium Dichloroacetate and Sodium Valproate on Infectious Inflammation Pathways

**DOI:** 10.3390/pharmaceutics15122715

**Published:** 2023-11-30

**Authors:** Donatas Stakišaitis, Linas Kapočius, Evelina Kilimaitė, Dovydas Gečys, Lina Šlekienė, Ingrida Balnytė, Jolita Palubinskienė, Vaiva Lesauskaitė

**Affiliations:** 1Department of Histology and Embryology, Medical Academy, Lithuanian University of Health Sciences, 44307 Kaunas, Lithuania; linas.kapocius@gmail.com (L.K.); lina.slekiene@lsmu.lt (L.Š.); ingrida.balnyte@lsmu.lt (I.B.); jolita.palubinskiene@lsmuni.lt (J.P.); 2Laboratory of Molecular Oncology, National Cancer Institute, 08660 Vilnius, Lithuania; 3Laboratory of Molecular Cardiology, Institute of Cardiology, Lithuanian University of Health Sciences, Sukileliu Ave., 50161 Kaunas, Lithuania; dovydas.gecys@lsmu.lt

**Keywords:** inflammation, viral infection, bacterial infection, thymus, thymocytes, investigational medicinal preparation, genes

## Abstract

The research presents data from a preclinical study on the anti-inflammatory effects of a sodium dichloroacetate and sodium valproate combination (DCA–VPA). The 2-week treatment with a DCA 100 mg/kg/day and VPA 150 mg/kg/day combination solution in drinking water’s effects on the thymus weight, its cortex/medulla ratio, Hassall’s corpuscles (HCs) number in the thymus medulla, and the expression of inflammatory and immune-response-related genes in thymocytes of male Balb/c mice were studied. Two groups of mice aged 6–7 weeks were investigated: a control (*n* = 12) and a DCA–VPA-treated group (*n* = 12). The treatment did not affect the body weight gain (*p* > 0.05), the thymus weight (*p* > 0.05), the cortical/medulla ratio (*p* > 0.05), or the number of HCs (*p* > 0.05). Treatment significantly increased the Slc5a8 gene expression by 2.1-fold (*p* < 0.05). Gene sequence analysis revealed a significant effect on the expression of inflammation-related genes in thymocytes by significantly altering the expression of several genes related to the cytokine activity pathway, the inflammatory response pathway, and the Il17 signaling pathway in thymocytes. Data suggest that DCA–VPA exerts an anti-inflammatory effect by inhibiting the inflammatory mechanisms in the mouse thymocytes.

## 1. Introduction

The development of new drugs and the search for new therapeutic indications for registered medicines treating severe infections or cancer is an important research area of medicine. This article presents data from a preclinical study on the anti-inflammatory effects of a combination sodium dichloroacetate (DCA) and sodium valproate (VPA) preparation (DCA–VPA). VPA and DCA are known medicines with long-standing therapeutic experiences and established safety profiles, dosages, and blood concentrations. They are attractive candidates for studies to elucidate potential new therapeutic indications.

Severe infections disrupt systemic metabolism [1,2,3], which can cause death [4,5]. Mitochondria play a role in activating antiviral and anti-inflammatory mechanisms [6]. DCA and VPA investigational drugs have anti-inflammatory effects on viral and bacterial infections [7,8]. DCA is an inhibitor of pyruvate dehydrogenase kinase (PDK), inhibiting PDK activity and increasing the activity of pyruvate dehydrogenase (PDH) and its complex (PDHC) [9,10]. DCA is commonly used to treat diseases associated with mitochondrial defects and increased congenital or acquired lactic acid production [11]. Various acquired metabolic and related immune dysfunction diseases are characterized by increased pathological expression of PDK, which inhibits PDHC, increases lactic acid production, causes its accumulation in tissues and blood, and releases inflammatory mediators [12,13]. PDK is a potential therapeutic target; its inhibition by DCA is meaningful in restoring immune and non-immune cell function [14,15,16]. PDH inactivation during infection is detected in peripheral blood mononuclear cells, skeletal muscle cells, vascular endothelial cells, and macrophages in severe infectious inflammation and sepsis [17,18]. The PDH subunit A1 (PDHA1)’s inactivation is associated with metabolic reprogramming and lactic acid overproduction [17,19]. Elevated blood lactic acid concentration is a biomarker of infection severity [12,20]. As a pro-inflammatory metabolite, lactic acid increases the production of pro-inflammatory cytokines [21,22]. Lactic acid, the end product of glycolysis, is an independent predictor of poor prognosis in sepsis, infectious coronavirus disease caused by SARS-CoV-2 virus (COVID-19), malaria, and other infections [23,24,25,26]. DCA regulates immune cells’ anabolic and catabolic energy supply [8,27] and increases resistance to infection [28].

VPA is an antiepileptic drug [29] also being investigated as an immunomodulator [30], and it has been studied to treat various viral infections [31]. VPA is a histone deacetylase (HDAC) inhibitor [32]. HDAC inhibitors may be anti-inflammatory drugs [33]. VPA is a multifunctional regulator of cells of the innate and adaptive immune system and reduces macrophage infiltration in various models of inflammation [7], significantly down-regulated pro-inflammatory genes and macrophage infiltration in the kidney [34], reduced induced neutrophil influx in a Balb/c mouse model of lung inflammation, and induced a reduction in neutrophil infiltration in bronchoalveolar fluid [35]. The reduced ability of immune cells to induce a pro-inflammatory response after VPA treatment may suggest new therapeutic options for managing septic shock in Balb/c mice [36]. VPA attenuated the progression of Coxsackie B3 virus-induced myocarditis and the mortality of male BALB/c mice [37]; and it inhibited leukocyte migration into the peritoneal cavity by 92% in a peritonitis-induced model in male rats, reducing TNF-α, Il1β, and Il6 levels with effects similar to indomethacin [38].

Pro-inflammatory immune cells derive most of their energy from aerobic glycolysis [39]. Glucose uptake and glycolysis are essential for the rapid growth and proliferation of virus-activated T cells [40,41]. Elevated glucose levels favor the progression of infection [42,43]. VPA and DCA reduce blood glucose levels in animals and humans [44,45,46].

VPA can activate Slc5a8, a sodium- and chloride-dependent monocarboxylate co-transporter encoded by the Slc5a8 gene (*Slc5a8*), u-regulating the expression of *Slc5a8* through DNA methylation [47,48]. Slc5a8 has a physiological function by transporting short-chain fatty acids into the cell and regulating mitochondrial metabolism; Slc5a8 is a transporter of DCA into the cell [49,50]. Slc5a8 regulates mitochondrial metabolism by participating in the mitochondrial β-oxidation pathway [51,52].

The thymus is the primary lymphoid organ where T lymphocytes mature and differentiate during the immune response. The thymus is responsible for generating T lymphocytes and is particularly vulnerable to infections [53]. Our previous research has shown that long-term monotherapy with DCA or VPA reduces the thymus weight of male Wistar rats [54,55].

This study aimed to investigate the effects of the DCA–VPA prolonged treatment of Balb/c mice on the thymus, its structure, and the expression of inflammation and immune-response-related genes in murine thymocytes. The objectives were to determine if the DCA–VPA treatment has an adverse effect on the mouse body and thymus weight, the effect on the expression of Hassall’s corpuscles in the thymus medulla, and the effects on the expression of Slc5A8 and the cytokine activity pathway, inflammatory response pathway, and Il17 signaling pathway genes in male Balb/c mice thymocytes.

The study results show that the treatment with DCA–VPA of Balb/c male mice results in a synergistic effect of the components of the investigational product, which is mediated by the VPA activation of DCA transport into thymocytes via the Slc5a8 co-transporter gene’s upregulation. This study has shown that DCA–VPA has an anti-inflammatory effect by inhibiting the expression of inflammation-promoting genes and increasing the expression of genes that inhibit infection-induced inflammation in murine thymocytes. This effect is related to inhibiting pro-inflammatory biological pathways. DCA–VPA is thus a suitable treatment option to reverse metabolic disturbances caused by inflammation.

## 2. Materials and Methods

### 2.1. The Investigational Medicinal Product

The investigational medicinal product is a combination of sodium dichloroacetate (DCA; Sigma-Aldrich, Steinheim, Germany) and sodium valproate (VPA; Sigma-Aldrich, Steinheim, Germany). The combination of these medicinal products (DCA–VPA) is patented by us as a new medicinal product for the treatment of cancer (official bulletin of the state patent bureau of the Republic of Lithuania, No. 6874, filling date 17 April 2020 https://vpb.lrv.lt/uploads/vpb/documents/files/VPB-OB-Nr23-2021-12-10-1d.pdf (accessed on 1 November 2023); a European patent application is submitted (European patent application no. 21168796.7, filing date 16 April 2021 https://register.epo.org/application?number=EP21168796 (accessed on 1 November 2023), as well as for the treatment of viral and bacterial infections (national patent application no. LT2023 532; 22 August 2023). Mice were treated with a drinking aqueous solution of the DCA–VPA combination (DCA 100 mg/kg and VPA 150 mg/kg/day) for two weeks. Considering the synergistic mechanism of action of DCA and VPA, the doses of DCA and VPA were half the doses we previously reported administered to animals treated with monotherapy [54,55].

### 2.2. The Groups of Mice and Experiments

The effect of DCA–VPA on the thymus gland of mice was investigated in two groups of male Balb/c mice aged 6–7 weeks: a control group (*n* = 12) and a matched group of male mice treated with a DCA–VPA solution (*n* = 12). The use of experimental animals for the studies was approved by the State Food and Veterinary Service of Lithuania (No. G2-198 of 16 March 2022). The animals were purchased from the Vivarium at the Veterinary Academy of the Lithuanian University of Health Sciences (Kaunas, Lithuania). The experiment was conducted at the Biological Research Center, Lithuanian University of Health Sciences (Kaunas, Lithuania). Animals were housed in standard colony cages with free access to food, warmth (21 ± 1 °C), humidity, and light/dark (12 h/12 h). Commercial pellet feed was provided ad libitum. The experiments followed laws and institutional animal care guidelines to avoid unnecessary animal suffering. After a 3-day acclimatization period, treatment of the animals with the DCA–VPA combination was initiated. The only source of drinking water was the DCA–VPA solution for the treated mice, while the control mice were supplied with fresh tap water; the DCA–VPA solution and water were available ad libitum to the animals. At the end of the experiment, 6 control and 6 treated mice were assigned to histological analysis of the thymus. Another 12 mice (6 control and 6 DCA–VPA-treated) were used to study *Slc5a8* expression, and RNR of 11 mice thymocytes (6 control and 5 DCA–VPA-treated) was used for the sequencing study (thymocyte RNR sequencing of 1 treated mouse was not tested due to insufficient RNA quality).

### 2.3. Thymus and Thymocyte Preparation

After the experiment, the animals were sacrificed in a 70% CO_2_ chamber. To minimize the thymus contamination with red blood cells, the carotid arteries and the aorta were cut, and the animals exsanguinated. Upon killing the animals, their thymus was harvested, and the contaminating blood was removed by rinsing with Dulbecco’s modified Eagle’s medium (DMEM) (ThermoFisher Scientific, Bleiswijk, The Netherlands) supplemented with 2 mM L-glutamine (ThermoFisher Scientific, Vilnius, Lithuania) and penicillin–streptomycin solution (ThermoFisher Scientific, Vilnius, Lithuania). The weight of the thymus was evaluated. Thymus glands were collected for histomorphometric evaluation.

For the evaluation of isolated thymocytes, the thymus was processed by removing the connective tissue surrounding the gland. The thymus was minced and heavily pipetted several times. More significant cellular accumulations and connective tissue were removed by passing the suspension through two layers of surgical gauze. The lymphocyte suspension was washed of red blood cells by centrifugation in DMEM twice for 8 min each. The supernatant was removed. The isolated thymocytes were resuspended in DMEM (Thermo Fisher Scientific, Bleiswijk, The Netherlands) supplemented with 2 mM L-glutamine (Thermo Fisher Scientific, Vilnius, Lithuania) and penicillin–streptomycin solution (Thermo Fisher Scientific, Vilnius, Lithuania).

### 2.4. The Histological and Immunohistochemical Examination of the Thymus

The thymus samples were fixed in 10% neutral-buffered formalin, dehydrated, and embedded in paraffin; sections of 3 μm were obtained and stained with hematoxylin and eosin (H&E) (Sigma-Aldrich, Darmstadt, Germany). Immunohistochemical staining was performed on slices placed on glass slides coated with poly-L-lysine. Deparaffinization of sections in xylene and rehydration were followed by pretreatment with the antigen-retrieval solution Tris/EDTA buffer, pH 9, in a 95 °C pressure cooker for 20 min. Incubation with cytokeratin monoclonal antibodies (clone 34βE12, dilution 1:100, Dako, Glostrup, Denmark) was used to identify high-molecular-weight cytokeratins (HMW CK). Each batch of slides was used for the positive control, and the negative control was achieved by omitting the primary antibody. The immunohistochemical reaction was revealed using an EnVision FLEX+, HRP kit (Dako, Glostrup, Denmark). EnVision FLEX Hematoxylin (Dako, Glostrup, Denmark) was used to counterstain the sections. The micro-morphometric and -morphologic evaluation of the samples was performed with an OLYMPUS BX53F microscope (Tokyo, Japan) equipped with a digital camera (Q-imaging, Surrey, BC, Canada) and using the Image-Pro Plus version 7.0 software (Media Cybernetics, Rockville, MD, USA). The thymus’ entire area and the cortical and medullary parts were measured, and the cortical/medullary parts ratio was calculated. Both lobes were scanned with the scanning microscope and measured using Image-Pro Plus version 7.0 software. The cortical and medullary parts were evaluated by measuring the total area (mm^2^) of each of the two lobes seen in one section. Hassall’s corpuscles (HCs) in the thymic medulla were assessed. Even a tiny cluster of epithelial reticular cells and a concentric arrangement of the cells was counted as an HC. Two independent histologists counted the number of HCs in the thymus medullary portion (the resulting number of HCs did not differ by more than 5% between the two investigators), and the results of the study are presented as the mean number of HCs per mm^2^ of the thymic area in each group.

### 2.5. Extraction of Total RNA from Thymocytes

A commercial TRIzol™ Plus RNA Purification Kit (Invitrogen™, Thermo Fisher Scientific, Bleiswijk, The Netherlands) was used to extract total RNA from the cells. Total RNA was quantified using a NanoDrop 2000 spectrophotometer (Thermo Scientific, Waltham, MA, USA). The integrity of the total RNA (RIN) was evaluated on an Agilent 2100 device (Agilent Technologies, Santa Clara, CA, USA) using an RNA 6000 Nano kit (Agilent Technologies, Santa Clara, CA, USA). Total RNA sample preparations were aliquoted and stored at −80 °C until subsequent downstream analysis.

### 2.6. Evaluation of the Slc5a8 Expression in Thymocytes

Total RNA was reverse transcribed using a High-Capacity cDNA Reverse Transcription Kit (Applied Biosystems™, Thermo Fisher Scientific, Bleiswijk, The Netherlands) supplemented with RNaseOUT™ Recombinant Ribonuclease Inhibitor (Invitrogen™, Thermo Fisher Scientific, Bleiswijk, The Netherlands). The real-time polymerase chain reaction (PCR) for *Slc5a8* (TaqMan Assay ID: mm00520629_m1) was performed using TaqMan chemistry (Applied Biosystems™, Thermo Fisher Scientific, Bleiswijk, The Netherlands), according to manufacturer’s instructions on a 7900 Real-Time PCR System (Applied Biosystems™ Thermo Scientific, Carlsbad, CA, USA). Samples were run in triplicate. The glyceraldehyde-3-phosphate dehydrogenase gene (*Gapdh*) was used as a reference (TaqMan Assay ID: mm99999915_g1). Gene expression alterations were analyzed using the 2^−∆∆Ct^ method [56].

### 2.7. Next-Generation Sequencing

According to the manufacturer’s instructions, next-generation sequencing (NGS) libraries were prepared using the QIAseq-targeted RNA Inflammation and Immunity Transcriptome panel kit (Qiagen, Hilden, Germany). Samples with RIN > 8.5 were used in sequencing analysis. Briefly, 400 ng of RNA was treated with DNAse I and reverse transcribed. Next, 20 ng of cDNA was used in the barcode assignment procedure, followed by magnetic bead purification. Barcoded cDNA was amplified during 8-cycle PCR and purified using magnetic beads. Next, 1st-stage barcoded PCR products were uniquely indexed during the 22-cycle 2nd-stage PCR step. Final PCR amplicons were purified using magnetic beads. NGS libraries were quantified using a Qubit™ High Sensitivity dsDNA Quantification Assay kit (Invitrogen™, Thermo Fisher Scientific, Bleiswijk, The Netherlands) on a Qubit 4.0 fluorometer (Invitrogen™, Thermo Fisher Scientific, Bleiswijk, The Netherlands). Size ranges of NGS libraries were determined using a High Sensitivity DNA 1000 kit (Agilent Technologies, Santa Clara, CA, USA) on the Agilent 2100 device. Libraries were appropriately denatured and diluted according to the NextSeq library denaturation and dilution guide. A sequencing run was performed on the Illumina NextSeq 550 sequencer (Illumina, San Diego, CA, USA) using 150-cycle Illumina High Output Kit v2.5 (Illumina, San Diego, CA, USA).

### 2.8. Statistical Analysis

Statistical analyses and graphical representations were performed using GraphPad Prism 9 software (GraphPad Software Inc., San Diego, CA, USA). The animal body weight, thymus weight data, thymic cortex/medulla areas ratio, and HC count are expressed as mean ± SD values. The ratio of the areas of the cortex and medulla in each individual of the group was calculated, and the mean value was obtained. The normality assumption of the thymus based on research and *Slc5a8* expression data distribution were evaluated and verified using the Shapiro–Wilk test. Quantitative differences between the two groups were evaluated using the Mann–Whitney *U* test. Results with a value of *p* < 0.05 were considered statistically significant.

### 2.9. Bioinformatic Analysis

The quality of the data was assessed using MultiQC v1.13 [57]. Adapter sequences with 3′ nucleotides and sequences shorter than 15 nucleotides, as well as sequences with a quality score lower than 25, were removed using Cutadapt v1.9.1 [58]. The human genome (GRCh38.p13) was downloaded from the Ensembl database [59]. The remaining sequences were aligned to the human genome using the STAR 2.1.3 tool [60]. The gene expression matrix was obtained using FeatureCounts v3.15 [61]. Sequence expression was normalized using the upper quartile method, and genes with a total expression across samples less than 50 were removed. Differential gene expression analysis was performed using DESeq2 v3.15 [62], and *p*-values were adjusted using the Benjamin–Hochberg method. Enrichment analysis of biological pathways was conducted using the DAVID server [63,64].

Data from treated cells were compared with untreated thymocytes (*n* = 6). Gene expression data from treated and control conditions were normalized and logarithmically transformed. Enrichment analysis results from Gene Set Enrichment Analysis (GSEA) were analyzed using the Enrichplot v1.2 and ClusterProfiler v4.8.2 R packages [65]. GSEA was performed using predefined algorithms to calculate enrichment scores and *p*-values for each dominant pathway or gene set. Gene set annotations were obtained from the Gene Ontology (GO), KEGG, and Reactome databases. The impact of DCA–VPA treatment was calculated by comparing gene expression in treated and control mouse cells. Significant changes in gene expression were determined when *p* < 0.05.

## 3. Results and Discussion

### 3.1. Effects of DCA–VPA Treatment on Body Weight, Thymus Structure, and Number of Hassall’s Corpuscles in the Medulla of the Thymus

The thymus is the central organ of the immune system, where T lymphocytes are formed, mature, and differentiate from thymocytes [66]. Under malnutrition or infection, mice show atrophy of the thymus, markedly reduced cortical tissue, a thymocyte count in the gland, and changes in chemokines and cytokines in the thymus tissue [53]. The mammalian thymus lobe structure is divided into the cortex and the medulla, with the cortex having a higher density of lymphocytes than the medulla. Immature CD4^−^CD8^−^ (double-negative; DN) and CD4^+^ CD8^+^ (double-positive; DP) thymocytes are located in the thymic cortex, whereas more mature CD4^+^ CD8^−^ or CD4^−^CD8^+^ single-positive thymocytes are located in the medulla [67]. It is, therefore, equally important in drug studies to determine the effect of the investigational medicine preparation on the weight of the animal and the weight and structural changes in the thymus.

This study found a significant increase in body weight over two weeks in the control animals (*n* = 12; 16.90 ± 2.33 g at the start and 20.68 ± 2.16 g at the end of the investigation; *p* < 0.0006) and the DCA–VPA-treated animal group (*n* = 12; 18.89 ± 1.30 g vs. 21.53 ± 1.33 g, respectively; *p* < 0.0001). DCA–VPA treatment did not affect body weight gain in the animals (*p* > 0.05). DCA–VPA treatment also did not affect thymus weight in mice (0.118 ± 0.011 g in the control and 0.118 ± 0.028 g in the treated group; *p* > 0.05). Previously, we reported that 4 weeks of monotherapy with DCA, at twice the dose of the DCA–VPA combination, caused a thymus weight decrease in DCA-treated gonad-intact male rats compared with their control, and no such impact was found in castrated DCA-treated males [54]. Also, VPA monotherapy at twice the dose of the DCA–VPA combination significantly reduced the thymus weight of castrated male Wistar rats after 4 weeks of treatment. However, it did not affect the thymus weight of intact males [68].

The thymic cortex/medulla ratio test can be used as an objective parameter to determine the effect of an investigational drug. Thymic atrophy and the associated decrease in the cortical/medulla ratio may be helpful as a quantitative indirect marker for screening histological examination [69]. The thymus contains a unique structure called Hassall’s corpuscles (HCs) found in the medulla. HCs appear as a collection of cells with a typical diameter of 20–100 μm [70]. The size of HCs correlates with the thymus size [71], making it difficult to detect in the mouse gland and for visualization requiring antigen-specific immunolabelling. Figure 1 shows immunohistochemical images of thymus specimens: histological specimens show the cortical and medullary areas and the HCs in the medulla.

Comparison of the thymus cortex and medulla histological structure of the control and the DCA–VPA-treated groups (*n* = 6 in each group) did not show a significant effect of 2 weeks of treatment. No DCA–VPA treatment effect was determined on the cortex-and-medulla ratio (3.30 ± 0.72 of control and 3.49 ± 1.25 of DCA–VPA-treated mice thymus; *p* > 0.05). The comparison of the number of HCs between the control group (9.70 ± 5.53 in mm^2^) and the DCA–VPA-treated group (8.18 ± 2.76 in mm^2^) revealed no significant difference (*p* > 0.05). It was reported that 4 weeks of treatment with high DCA doses caused a significant increase in the number of HCs in gonad-intact males but not in castrated rats [54].

The structure of HCs is the final stage of differentiation of the Aire+ lineage [72,73]. The size and number of HCs decrease with gland involution [74]. Human studies have shown that the AIRE gene is located on chromosome 21: the trisomy of chromosome 21 in patients (Down syndrome) results in a greater size and number of HCs in the thymus [75]. The formation of HCs is thought to be linked to an aging atrophy phenotype with reduced IFNα production, suggesting that the lack of an inflammatory signal leads to impaired thymocyte development [76]. Confirmation that medullary thymic epithelial cells (mTECs) and HCs are involved in the induction of a pro-inflammatory microenvironment was obtained by studying Aire KO mice, which are depleted of post-Aire cells, and HCs are nearly missing in the Aire KO mouse [77] and have reduced expression of inflammatory mediators [76]. HCs may be necessary for the lymphoid tissue microenvironment’s specific tonic inflammatory signaling. HCs express thymic stromal lymphopoietin, which can induce normal T cells to Treg [78,79]. The evidence of mTECs and HCs as inducers of tonic pro-inflammatory microenvironments in the thymus [80] highlights the importance of studying the effects of investigational drugs on the expression of genes involved in thymocytes in inflammation and the immune response.

### 3.2. The Effect of DCA–VPA Treatment on Thymocytes’ Slc5a8 Gene Expression

Table 1 shows the effect of the DCA–VPA on *Slc5a8* RNR expression in the thymocytes of Balb/c male mice in vivo.

Two weeks of treatment with the combination of DCA 100 mg/kg per day and VPA 150 mg/kg per day results in a significant upregulation of *Slc5a8* by 2.1-fold in thymocytes of Balb/c male mice. Figure 2 shows the effect of DCA–VPA treatment on Slc5a8 co-transporter gene expression in the thymocytes.

Compared to the control, the combination treatment resulted in a significant increase in *Slc5a8* expression in thymocytes; thus, the synergistic mechanism is manifested by VPA, improving the cellular transport of DCA and thus enhancing the pharmacological effects of DCA. Higher co-transporter expression may lead to better substrate accumulation in the cell. Slc5a8 is a sodium- and chloride ion-dependent and sodium-linked monocarboxylate co-transporter that transports the dichloroacetate anion into cells [81,82]. Studies in male wild-type c/ebpδ^+/−^ and c/ebpδ^−/−^ mice indicate that Slc5a8 has a physiological role in the transport of short-chain fatty acids as a substrate for the β-oxidation pathway in mitochondria into the cell. Upregulated Slc5a8 is essential in regulating mice’s cell metabolism and inflammation mechanisms [83]. The importance of Slc5a8 for immune homeostasis and inhibition of inflammation is highlighted. Short-chain fatty acids transported to immune cells via Slc5a8 alter HDAC activity and have immunomodulatory effects, such as blocking the development of dendritic cells, releasing cytokines, and inducing T-cell apoptosis [81,83]. *Slc5a8* expression can be upregulated by DNA demethylation. VPA can induce the effect on Slc5c8 gene expression upregulation via DNA demethylation in cancer cells [81,84].

### 3.3. The Effect of DCA–VPA on Related Inflammation and Immune Response Genes’ Expression in Mice Thymocytes

It was reported that the human and mouse thymus is a distinct organ that produces pro-inflammatory mediators even under physiological conditions [85,86]. Therefore, with its constant microenvironment tonic inflammation background, the thymus can be a used as a model for studying the effects of investigational drugs that suppress inflammation and the immune response through changes in the expression of inflammation- and immune-response-related genes in thymocytes.

Table 2 shows the mean and log_2_ data for the expression of genes related to inflammation and the immune response in thymocytes in the control and DCA–VPA-treated groups and the log_2_ change comparing the control and treated groups. Genes with significantly increased expression after DCA–VPA treatment are shown in green, and genes with significantly decreased expression after treatment are shown in yellow.

When comparing the changes in genes related to inflammation and the immune response in control and treated male mice, 19 genes were significantly decreased in the thymocytes of the mice treated for two weeks with the DCA–VPA combination, i.e., *Ccl2*, *Ccl3*, *Ccr5*, *Cdkn1a*, *Cebpb*, *Csf3*, *Cxcl1*, *Cxcl2*, *Cxcl3*, *Il17f*, *Il18r1*, *Il1rl1*, *Il6*, *Ptgs2*, *Rel*, *Spp1*, *Thbs1*, *Tnfsf11*, and *Vegfa*, and there was a significant increase in the expression of 20 genes, i.e., *Bmp4*, *Bmp7*, *C3*, *Ccl19*, *Ccl25*, *Ciita*, *Cmklr1*, *Cx3cl1*, *Cxcl11*, *Cxcl12*, *Cxcl13*, *Il2*, *Il7*, *Il21*, *Il23a*, *Il27*, *Il33*, Kitl, *Nos2*, and *Pparg*. Figure 3 shows data on the effect of DCA–VPA treatment on the expression of genes significantly downregulated or upregulated in male mice thymocytes.

#### 3.3.1. A Possible DCA–VPA Treatment Effect on Thymus Function

In the context of the development of new drugs that inhibit inflammation and the immune response, it is essential to identify their effects on the integrated function of the thymus, possible adverse effects on T-lymphocyte lymphopoiesis, and changes in the signals to the T-cell progenitors from the thymus.

Compared to controls, a two-week treatment of mice with DCA–VPA increased the expression of *Cxcl12*, *Il2,* and *Il7* in thymus cells and inhibited the expression of *Il6* and *Ccl25* (Figure 3). Cxcl12 promotes cell proliferation and differentiation to DN4 and the DP stage [87]. Cxcl12 is linked to DP cell migration and its receptor Cxcr4 is highly expressed in this cell population [88]. Cxcl12 binds to the Cxcr4 receptor and attracts immature CD4^−^CD8^−^ and CD4^+^CD8^+^ cells, and all thymocyte subsets are responsive to Ccl25, while its Ccr9 receptor is expressed in all stages of thymocyte differentiation in mice (five-fold). Since Ccr9 is reduced in mature thymocytes and Ccl25 sensitivity is lost just before thymocyte emigration, the Ccl25/Ccr9 interaction appears vital in thymic cell retention [89]. Thus, the effect of DCA–VPA, which may be related to the release of T cells from the thymus, cannot be excluded. Thymocyte differentiation and the release of mature T lymphocytes depend on the tissue microenvironmental cytokines Il2, Il6, and Il7 [66]. Il2 is a pro-inflammatory cytokine, and its increase due to DCA–VPA exposure could be considered a positive effect, as it has been reported that thymocytes from acutely infected mice with *T. cruzi* have a reduced response to mitogens due to reduced secretion of Il2 [90]. Il6 can promote mitogen-induced thymocyte proliferation, associated with its anti-apoptotic function; Il6 induces CD8^+^ T-cell proliferation in synergy with Il7 [91]. In mice, Il7 production is associated with improved thymic cell health and functionality: Il7 is involved in normal T-lymphocyte development, stimulates survival and expansion of immature thymocytes, and increases thymocyte numbers [92,93].

#### 3.3.2. Effects of DCA–VPA Treatment on Cytokine Activity Pathway, Inflammatory Response Pathway, and Il17 Signaling Pathway Genes in Male Balb/c Mice Thymocytes

The differently expressed genes between control and DCA–VPA-treated thymic cells are enriched in the cytokine activity, inflammatory response pathway, and Il17 signaling pathway, as demonstrated by GO and KEGG analysis. Table 3 shows the genes involved in the biological pathways of cytokine activity, inflammatory response, and Il17 signaling in thymocytes, and the effects of DCA–VPA treatment on genes.

##### DCA–VPA Effects on the Cytokine Activity Pathway

Sequencing analysis showed that the treatment of male mice with DCA–VPA had a significant effect on 25 genes of the cytokine activation pathway: 11 genes’ expression was significantly decreased, and 14 genes’ expression was significantly increased by treatment (Figure 2, Table 2 and Table 3). Compared to the control, in the DCA–VPA-treated thymocytes group, decreased *Ccl2*, *Ccl3*, *Csf3*, *Cxcl1*, *Cxcl2*, *Cxcl3*, *Il17f*, *Il6*, *Spp1*, *Tnfsf11*, and *Vegfa* gene expression was found. Such test drug effects on inflammation and immune-response-related genes could be seen as suppressing inflammation and the immune response. Ccl2 and Ccl3 chemokines are classified as pro-inflammatory: Ccl2 stimulates chemotaxis of monocytes and a few cellular events associated with chemotaxis: Ccl3 function, macrophage–NK migration, and T-cell/DC interaction [94]. Cxcl1, Cxcl2, and Cxcl3 chemokines’ function is neutrophil trafficking [94]. Cx3cl1 is expressed mainly in immune cells, including monocytes and T cells, and deficiency of Cx3cr1 promotes pro-inflammatory cytokine production in macrophages and T cells [95]. Cx3cl1 acts via the Cx3cl1/Cx3cr1 axis: the chemokine receptor Cx3cr1 is a known marker of anti-inflammatory monocytes, providing a pro-survival signal to anti-inflammatory monocytes; it also presents in NK cells and T cells, where it mediates proliferation, migration, and adhesion [94,96]. Helper T cells (Th17) are potent inducers of tissue inflammation by releasing Il17f [97]. Il6 plays an important role during the early immune response to infection, causing B lymphocytes to differentiate into mature, immunoglobulin-secreting plasma B cells; signaling also initiates T-cell activation, growth, and differentiation [98]. Spp1 at the cellular level is expressed in macrophages, dendritic cells, lymphoid cells, and mononuclear cells of the immune system [99]: severe COVID-19 disease is associated with macrophage SPP1 production, and SPP1 leads to the strong inflammatory response characteristic of severe COVID-19 disease [100]. Reduced gene expression of *Tnfsf11* in thymocytes is associated with the differentiation of mature thymocytes into Tregs and cell release [101]; Tregs suppress the immune response, maintain homeostasis, and may inhibit T-cell proliferation and cytokine production [102]. Vegf and its receptor’s upregulation is linked to the pathophysiology of several human viral diseases [103].

The effect of DCA–VPA on the reduction in Csf3 expression needs to be addressed separately. Csf3 (granulocyte colony-stimulating factor; G-CSF) is an essential cell growth factor that supports neutrophil progenitor cell proliferation, survival, and differentiation and is a potent immune regulator of T cells [104]. G-csf is released from lung cells in response to the pro-inflammatory cytokines TNF-α and Il1β [105]. G-csf receptor (G-csfr) blockade reduces neutrophil infiltration and neutrophil-mediated inflammation in mouse models of infection and asthma [106,107]. The blockade of G-CSF or its receptor G-CSFR could be a treatment strategy for such patients. A case report describing the onset of ARDS in five patients in whom G-CSF was administered in combination with chemotherapy or hematopoietic cell transplantation has been published [108]. G-CSF administration may worsen lung function, especially in cytokine release syndrome, sepsis, or ARDS, which are also critical features of COVID-19 disease [105].

Compared to the control, in the DCA–VPA-treated thymocytes group, increased *Bmp4*, *Bmp7*, *Ccl19*, *Ccl25*, *Cx3cl1*, *Cxcl12*, *Cxcl13*, *Il2*, *Il7*, *Il21*, *Il23a*, *Il27*, *Il33*, and *Kitl* gene expression was found in mice thymocytes. Bmp4 molecules were found to act as a host response against SARS-CoV-2 [109]. Studies have shown that Bmp7 is an anti-inflammatory factor that downregulates basal and TNFα-stimulated production of the pro-inflammatory cytokines Il6, Il8, and Il1β, along with the chemokine Ccl2, and induces M2 macrophage differentiation [110,111] Ccl19 and Ccl25 chemokines’ function is as homeostatic thymocyte migration induction: Ccl19 is responsible for T-cell and dendritic cell homing to lymph node, and Ccl25 is responsible for T-cell homing to the gut [94]. The Cxcl12 chemokine is responsible for bone marrow homing, while Cxcl13 is responsible for B-cell positioning in lymph nodes [94]. Thus, the DCA–VPA treatment is related to an anti-inflammatory effect. Il2 and Il21 have specific functions during T-cell differentiation and homeostasis [112], while Th17 are potent inducers of tissue inflammation through the release of Il21 [97]. However, our study shows that DCA–VPA inhibits the Th17 pathway. Il7 regulates immature and mature T-lymphocyte homeostasis [113]. Il23a plays a role in inflammation, the immune response, and cell differentiation and survival [98]. Il27 is involved in helper T-cell differentiation [114]. Il33 is pro-inflammatory [98]. Kitl is a thymopoietic molecule [115]. In Il33 knockout mice, it was discovered that nuclear Il33 is associated with wound healing, as mice without the protein healed significantly slower than mice with the Il33 protein [116]. The functions of the above-mentioned related cytokine activity pathway genes suggest that DCA–VPA has anti-inflammatory and immunosuppressive properties by inhibiting the cytokine activity pathway.

##### DCA–VPA Effects on the Inflammatory Response Pathway

The DCA–VPA treatment significantly affected 25 genes of the inflammatory response pathway: 13 genes overlap with the cytokine response pathway, 7 genes’ expression is decreased (*Ccl2*, *Ccl3*, *Cxcl1*, *Cxcl2*, *Cxcl3*, *Il6*, *Il17f*), and 7 are upregulated (*Ccl19*, *Ccl25*, *Cx3cl1*, *Cxcl11*, *Cxcl13*, *Il23a*, *Il27*). Treatment with the investigational drug had an anti-inflammatory effect by significantly suppressing *Ccr5*, *Il1rl*, *Ptgs2*, *Rel*, and *Thbs1* expression. Ccr5 (chemokine receptor 5) is a common receptor for the chemotactic mediators Ccl3, Ccl4, and Ccl5. Ccr5 is expressed in many inflammatory cell lines, including T cells. The expression of Ccr5 is upregulated in models of inflammation, and Ccr5 antagonism may be a promising therapeutic strategy [117]. As indicated above, DCA–VPA increased *Il33* expression in Balb/c thymocytes but inhibited Il1rl (St2) receptor expression. Il33 targets are type 1 and type 2 immune cells. The Il1rl1 receptor is essential for the mechanisms underlying Il33 inflammation. Il33 is released after cell damage or tissue injury and activates signaling pathways in cells harboring the I1rl1l receptor. Thus, DCA–VPA could indirectly inhibit Il33-induced mechanisms by inhibiting *Il1rl1* gene expression. Mouse models support an essential Il33/St2 signaling role in allergic inflammatory mechanisms (in the nasopharynx, lung, and skin tissues). Il33/St2 is also important for non-allergic inflammatory pathways [118]. Ptgs2 (prostaglandin-endoperoxide synthase 2; Cox2) is a pro-inflammatory cyclooxygenase whose activity increases during inflammation. Cox2 is important for the production of pro-inflammatory eicosanoids. Cox2 gene expression is induced by the cytokines Il1β and TNFα [119]. Rel is a member of the NF-κB/Rel family of transcription factors restricted by hemopoietic cells and important for T-lymphocyte function. Rel^−/−^ mice had a significantly reduced T-cell proliferative response to the virus and attenuated local and systemic influenza virus-specific antibody responses, with normal levels of virus-specific cytotoxic T lymphocytes, which were able to clear the virus from the mouse lungs [120]. Thbs1 (thrombospondin 1) is released and increased during the acute phase of inflammation and plays a synergistic role in the inflammatory process; Thbs1 is a pro-inflammatory factor [120], which was increased among COVID-19 patients [121].

The upregulation effect of DCA–VPA treatment on inflammatory response pathway genes was found to overlap seven genes with the cytokine response pathway (*Ccl19*, *Ccl25*, *Cx3cl1*, *Cxcl11*, *Cxcl13*, *Il23a*, *Il27*). Additionally, the expression of five genes (*C3*, *Ciita*, *Cmklr1*, *Nos2*, *Pparg*) was found to be significantly increased by the treatment with DCA–VPA in thymocytes, which is also linked to the anti-inflammatory effects of test preparation. C3 protein is mainly located in thymic corpuscles, which may be involved in immune regulation [122]. C3 plays a crucial role in enhancing the T-cell response to bacterial infection by promoting the proliferation of antigen-exposed CD8 T cells [123]. Studies in mice show that C3 produced by the liver and lungs ensures the host’s response to infection by stimulating immune resistance. C3 alleviates severe bacterial pneumonia: intracellular complement protein modulates cell survival and provides a putative mechanism by which lung-derived C3 protects from tissue damage induced by pneumonia [124]. Ciita is a key transactivator of major histocompatibility complex II expression and controls antigen presentation followed by T-cell activation [125]. The epigenetic regulation of *Ciita* by VPA is the suspected cause of the increased expression of *Ciita* in thymocytes due to demethylation after therapy with DCA–VPA. If this is the case, such an effect may be beneficial for cancer therapy, as epigenetic inhibition of *Ciita* in tumor cells may prevent the immune system from recognizing the tumor antigen and, thus, the anti-tumor immune response [126].

The significance of Cmklr1 (chemokine-like receptor-1) is not fully understood [127,128]. Cmklr1 deficiency reduces leukocyte recruitment to inflammation-damaged lungs [127]. On the other hand, activation of Cmklr1 in a murine model of virus-induced pneumonia improved viral clearance and antiviral antibody production, reduced the expression of pro-inflammatory mediators, reduced complications, and improved mouse survival [129,130]. While clinical data are still lacking, preclinical data suggest that Cmklr1 may be a lung inflammation biomarker in pneumonia patients [128]. The enzyme NOS2 (nitric oxide synthase 2) and reactive oxygen species (ROS) are players in inflammatory and immune responses. However, the functional significance of the link between NOS2 and ROS during infection remains unclear [131]. It cannot be excluded that the possible increase in NOS2 expression may be due to the generation of ROS induced by DCA and VPA [132,133]. Peroxisome proliferator-activated receptor-γ (Pparg) is involved in anti-inflammatory processes through immune cells and is in anti-inflammatory macrophages [134].

##### DCA–VPA Effects on the Il17 Signaling Pathway

DCA–VPA treatment reduced the expression of *Ccl2*, *Cebpb*, *Csf3*, *Cxcl1*, *Cxcl2*, *Cxcl3*, *Il6*, *Il17f*, and *Ptgs2* genes, which are involved in the Il17 signaling in the thymocytes of male Balb/c mice. These genes are also involved in cytokine activity and inflammatory response pathways (Table 2). Il17 plays a key role in the cytokine storm, in ARDS, and is associated with alveolar inflammation and poor prognosis [135]. In mouse models, direct blockade of Il17 reduced lung injury [136]. In severe cases of COVID-19, elevated levels of Il17 signaling pathway pro-inflammatory chemokines and cytokines are directly correlated with increased severity of lung injury and prognosis [137].

Inflammation is a defensive cellular response to pathogens, infection, or tissue damage. It coordinates the interactions between different immune cells, a complex chain of molecular signaling mechanisms, the release of inflammatory mediators (chemokines, cytokines, etc.), and their effects on tissues. The novelty of this preclinical study is that DCA–VPA treatment suppresses the inflammatory response in thymocytes. This finding is consistent with previous studies showing the anti-inflammatory properties of both DCA and VPA in monotherapy. The synergism between DCA and VPA in activating DCA transport into the cell via Slc5a8 allows a reduction in the DCA dose.

A limitation of this study could be that while the thymus cell suspension used was composed of an absolute majority of T cells, a minority of other microenvironmental cells, such as thymic epithelial cells, dendritic cells, macrophages, and fibroblasts were not separated [105], which may have influenced the gene expression data to some extent. Also, for synergism, an additional justification, besides the effect of the combination on *Slc5a8*, would be a comparison with an untreated control, monotherapy, and treatment with combination. Another limitation of the study is that only male Balb/c mice were tested, so the results presented may not be consistent with other populations and do not reveal sex differences as they were not compared with similar data from female mice. However, the preclinical study data obtained are undoubtedly relevant for evaluating the anti-inflammatory effects of preparation, which are essential for further studies on the pharmacological efficacy of the DCA–VPA.

## 4. Conclusions

The treatment with the DCA–VPA combination did not affect the weight of the Balb/c male mice’s thymus and its structure. DCA–VPA significantly increased Slc5A8 gene expression in thymocytes, suggesting that VPA as an epigenetic demethylating agent may improve the intracellular delivery of DCA. Thymocytes are enriched in the cytokine activity pathway, inflammatory response pathway, and Il17 signaling pathway. DCA–VPA exhibits anti-inflammatory effects by inhibiting the inflammatory mechanisms of the pathways mentioned above by suppressing the expression of pro-inflammatory genes and activation of anti-inflammatory chemokines’ and cytokines’ gene expression in the thymocytes of male mice.

## 5. Patent

Medicinal product DCA–VPA is patented as a new medicinal product for the treatment of viral and bacterial infections (national patent application no. LT2023 532; 22 August 2023).

## Figures and Tables

**Figure 1 pharmaceutics-15-02715-f001:**
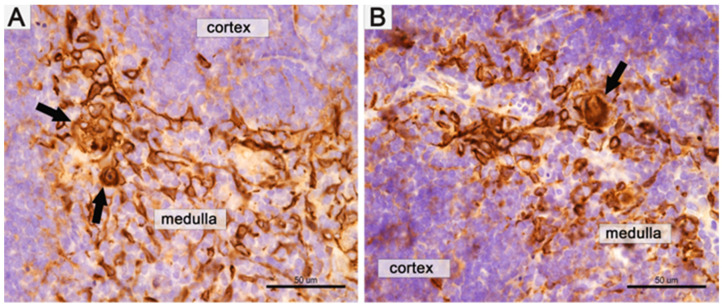
Immunohistochemical images of the Balb/c male mouse thymus. (**A**)—Thymus of the control group mouse (the cortex and medulla). (**B**)—Image of the DCA–VPA-treated mouse preparation. Hassall’s corpuscles (arrows) in the medulla. Thymic epithelial cells and HCs are positive for high-molecular-weight cytokeratins (clone 34βE12). Scale bar of (**A**,**B**)—50 μm.

**Figure 2 pharmaceutics-15-02715-f002:**
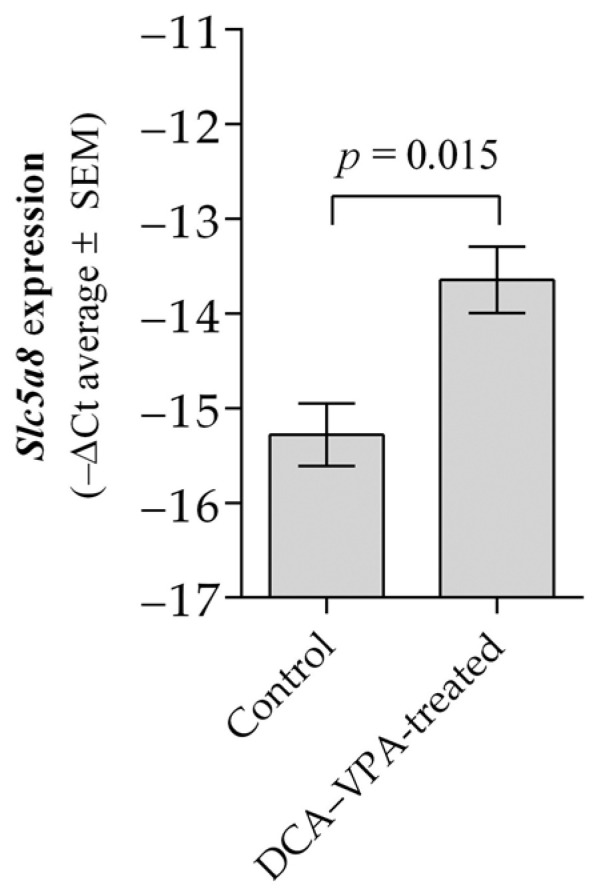
Slc5a8 co-transporter gene expression changes after treatment with DCA–VPA.

**Figure 3 pharmaceutics-15-02715-f003:**
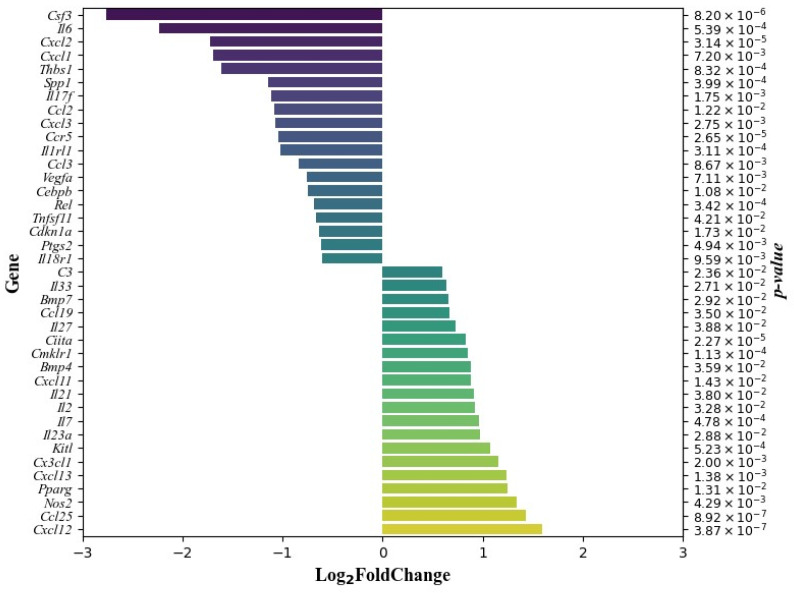
Genes whose expression in mouse thymocytes was significantly changed by DCA–VPA treatment. The data are expressed as Log_2_ fold change—genes with reduced expression are shown in descending order of repression, and genes with upregulated gene expression are shown in ascending order of repression.

**Table 1 pharmaceutics-15-02715-t001:** *Slc5a8* expression in male mouse thymocytes of the studied groups.

Groups Studied	*n*	Ct Mean		ΔCt Mean ± SEM	ΔΔCT
		*Slc5a8*	*Gapdh*		
Control	6	34.77	19.49	15.27 ± 0.329	−1.63
DCA–VPA-treated	6	33.40	19.76	13.64 ± 0.351 ^a^	

^a^ *p* = 0.015, compared with the control.

**Table 2 pharmaceutics-15-02715-t002:** Expression of genes related to inflammation and immune response in male mice thymocytes in the control and DCA–VPA-treated groups.

Gene	Log_2_ Fold Change	Gene Expression (Average)		Gene Expression (Log_2_)		*p* Value
		Treated	Control	Treated	Control	
*Bmp4*	0.878	108.197	58.504	6.758	5.870	0.036
*Bmp7*	0.661	212.423	134.572	7.731	7.072	0.029
*C3*	0.598	7367.336	4867.781	12.847	12.249	0.024
*Ccl19*	0.667	260.248	163.851	8.024	7.356	0.035
*Ccl25*	1.427	28,348.999	10,540.968	14.791	13.364	8.92 × 10^−7^
*Ciita*	0.828	1867.537	1052.728	10.867	10.040	2.27 × 10^−5^
*Cmklr1*	0.848	653.655	363.705	9.352	8.507	0.0001
*Cx3cl1*	1.156	708.182	317.965	9.468	8.313	0.0020
*Cxcl11*	0.883	1184.140	641.946	10.210	9.326	0.0143
*Cxcl12*	1.593	10,218.765	3386.550	13.319	11.726	3.87 × 10^−7^
*Cxcl13*	1.242	1620.704	685.156	10.662	9.420	0.0014
*Il2*	0.920	63.683	33.400	5.993	5.062	0.033
*Il7*	0.966	2624.135	1343.731	11.358	10.392	0.0005
*Il21*	0.912	30.857	16.548	4.948	4.049	0.038
*Il23a*	0.976	118.014	59.799	6.883	5.902	0.029
*Il27*	0.727	82.518	49.879	6.367	5.640	0.039
*Il33*	0.639	645.356	414.507	9.334	8.695	0.027
*Kitl*	1.077	566.866	268.273	9.147	8.068	0.0005
*Nos2*	1.337	136.272	54.373	7.090	5.765	0.0043
*Pparg*	1.246	879.010	370.480	9.780	8.533	0.013
*Ccl2*	−1.081	949.215	2007.774	9.891	10.971	0.012
*Ccl3*	−0.844	1822.248	3270.595	10.832	11.675	0.009
*Ccr5*	−1.041	552.400	1135.583	9.110	10.149	2.65 × 10^−5^
*Cdkn1a*	−0.639	2633.161	4101.501	11.363	12.002	0.017
*Cebpb*	−0.746	12,316.158	20,655.910	13.588	14.334	0.005
*Csf3*	−2.767	14.791	98.700	3.887	6.625	8.20 × 10^−6^
*Cxcl1*	−1.694	1174.780	3801.410	10.198	11.892	0.007
*Cxcl2*	−1.723	719.590	2373.510	9.491	11.213	3.14 × 10^−5^
*Cxcl3*	−1.073	150.984	317.559	7.238	8.311	0.0028
*Il1rl1*	−1.021	79.044	159.541	6.305	7.318	0.0003
*Il6*	−2.238	678.207	3198.631	9.406	11.643	0.0005
*Il17f*	−1.119	49.106	106.128	5.618	6.730	0.0017
*Il18r1*	−0.608	1246.327	1898.444	10.283	10.891	0.01
*Rel*	−0.691	3225.337	5204.208	11.655	12.345	0.0003
*Ptgs2*	−0.611	878.529	1340.712	9.779	10.389	0.005
*Spp1*	−1.142	343.025	757.012	8.422	9.564	0.0004
*Tnfsf11*	−0.669	2933.322	4664.196	11.518	12.187	0.042
*Thbs1*	−1.616	309.798	948.572	8.275	9.890	0.0008
*Vegfa*	−0.753	3633.337	6122.907	11.827	12.580	0.007

**Table 3 pharmaceutics-15-02715-t003:** Effects of DCA–VPA treatment on male mouse thymocyte genes for cytokine activity, inflammatory response, and Il17 signaling pathways.

Category and Term of Pathway	Count of Genes	*p* Value	Gene Expression	Fold Enrichment	*p* Value by Benjamini–Hochberg Method
GOTERM_ MF_DIRECT—GO:0005125~ cytokine activity	25	7.15 × 10^−30^	Decreased: *Ccl2*, *Ccl3*, *Csf3*, *Cxcl1*, *Cxcl2*, *Cxcl3*, *Il17f*, *Il6*, *Spp1*, *Tnfsf11*, *Vegfa* Increased: *Bmp4*, *Bmp7*, *Ccl19*, *Ccl25*, *Cx3cl1*, *Cxcl12*, *Cxcl13*, *Il2*, *Il7*, *Il21*, *Il23a*, *Il27*, *Il33*, *Kitl*	31.10	1.75 × 10^−27^
GOTERM_ BP_DIRECT—GO:0006954~ inflammatory response	25	2.01 × 10^−27^	Decreased: *Ccl2*, *Ccl3*, *Ccr5*, *Cxcl1*, *Cxcl2*, *Cxcl3*, *Il1rl1*, *Il6*, *Il17f*, *Il18r1*, *Ptgs2*, *Rel*, *Thbs1* Increased: *C3*, *Ccl19*, *Ccl25*, *Ciita*, *Cmklr1*, *Cx3cl1*, *Cxcl11*, *Cxcl13*, *Il23a*, *Il27*, *Nos2*, *Pparg*	20.20	2.81 × 10^−24^
KEGG_PATHWAY—hsa04657: IL-17 signaling pathway	9	7.94 × 10^−8^	Decreased: *Ccl2*, *Cebpb*, *Csf3*, *Cxcl1*, *Cxcl2*, *Cxcl3*, *Il6*, *Il17f*, *Ptgs2*	15.40	2.16 × 10^−6^

## Data Availability

The data presented in this study are available on request from the corresponding author.

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
