# Peer review of "Preclinical Study in Mouse Thymus and Thymocytes: Effects of Treatment with a Combination of Sodium Dichloroacetate and Sodium Valproate on Infectious Inflammation Pathways"

_pharmaceutics, 2023, doi:10.3390/pharmaceutics15122715_

Round 1

Reviewer 1 Report

Comments and Suggestions for Authors

The manuscript entitled, ‘The Effects of Combination Therapy of Sodium Dichloroacetate and Sodium Valproate on the Management of Infection Pathways: Pre-clinical Research in Murine Thymus and Thymocytes’ examines the anti-inflammatory effects of a sodium dichloroacetate and sodium valproate combination (DCA–VPA). Furthermore, the authors have studied the expression of inflammation-related genes in thymocytes. However, some major concerns should be addressed by the authors before any possible consideration of this manuscript to be published in the Journal of pharmaceutics.

1.      The aim and objectives of the study should be clearly stated in the last paragraph of the #Intriduction section.

2.      On what basis have the authors decided the concentration of treatments (100 mg/kg and 150 mg/kg) of DCA and VPA?

3.      As mentioned in #lines_265-268, why were significant differences in the body weight noticed in the control animals compared to DCA-VPA-treated animals?

4.      How did the DCA–VPA treatment significantly increase Slc5A8 gene expression in thymocytes? The authors must provide a detailed discussion in the respective section.

5.      The limitations of this work should be clearly stated in the #Conclusion section.

6.      The study fails to justify the word ‘the Management of Infection Pathways’ (mentioned in the #Introduction section), so the authors may think of another title representing the comprehensiveness of this study.

7.      The abbreviated words should be explained at their first use in the manuscript and then used consistently.

8.       The manuscript should be thoroughly checked for English language and grammatical errors.

Comments on the Quality of English Language

Moderate editing of English is required.

Author Response

Responses to the 1st Reviewer's comments

 Comments and Suggestions for Authors

The manuscript entitled, ‘The Effects of Combination Therapy of Sodium Dichloroacetate and Sodium Valproate on the Management of Infection Pathways: Pre-clinical Research in Murine Thymus and Thymocytes’ examines the anti-inflammatory effects of a sodium dichloroacetate and sodium valproate combination (DCA–VPA). Furthermore, the authors have studied the expression of inflammation-related genes in thymocytes. However, some major concerns should be addressed by the authors before any possible consideration of this manuscript to be published in the Journal of pharmaceutics.

Below are the corrections and responses to the Reviewer's comments in order.

  1. The aim and objectives of the study should be clearly stated in the last paragraph of the #Intriduction section.

Answer

Corrections were made in line with the comment. Text corrected in the Introduction section – new text added:” The study aimed to investigate the effects of the DCA–VPA prolonged treatment of Balb/c mice on the thymus, its structure, the effects on the expression of inflammation and immune response-related genes in murine thymocytes. The objectives were to determine if the DCA–VPA treatment has an adverse effect on mice body and thymus weight, to determine the effect on the expression of Hassall's corpuscles in the thymus medulla, the effects on the expression of Slc5A8 and cytokine activity pathway, inflammatory response pathway, Il17 signaling pathway genes in male Balb/c mice thymocytes. In the corrected text, lines 85–90.

 2. On what basis have the authors decided the concentration of treatments (100 mg/kg and 150 mg/kg) of DCA and VPA?

Answer

The concentrations of DCA or VPA in the solution used in our study are chosen from the concentrations of oral solution used by other researchers and in our previous studies. Only in this study, we used half the DCA or VPA concentrations because our studies have shown that the efficacy of DCA is increased in combination, i.e., the efficacy of half the dose of DCA in combination with VPA (e.g., for anti-cancer effects) is equivalent or even more significant for twice the dose of DCA. The synergism is what we have shown in the product patent. In the text under '2.1. Investigational medicinal product' is indicated: 'Mice were treated with a drinking aqueous solution of the DCA–VPA combination (DCA 100 mg/kg and VPA 150 mg/kg/day) for two weeks. Considering the synergistic mechanism of action of DCA and VPA, the doses of DCA and VPA were half the doses we have previously reported administered to rats treated with monotherapy [54,55]'. Text corrected to replace 'animals' with 'rats'. We consider that the information provided in the text is sufficient.

  1. As mentioned in #lines_265-268, why were significant differences in the body weight noticed in the control animals compared to DCA-VPA-treated animals?

Answer

DCA–VPA treatment did not affect on body weight gain in mice. We believe that the unclear wording may have led to a misunderstanding. We have corrected the sentence by deleting the words "Compared to the control". See lines 250-253 of the corrected text with corrections.

 4. How did the DCA–VPA treatment significantly increase Slc5A8 gene expression in thymocytes? The authors must provide a detailed discussion in the respective section.

Answer

Why we investigated Slc5A8 gene expression is stated in the 'Introduction' section (lines 78-83 of the corrected text). We also corrected this section to take account of the comment. We have also made additions to the text in lines 320-323. The corrections are in line with the Reviewer’s note.  

  1. The limitations of this work should be clearly stated in the #Conclusion section.

Answer

The limitation of the study is identified in lines 505-507 of the text. We believe it is sufficient to highlight the limitations in the 'Discussion' section. To increase the visibility of this statement, we have started it from a new line. This correction is sufficient.  

  1. The study fails to justify the word ‘the Management of Infection Pathways’ (mentioned in the #Introduction section), so the authors may think of another title representing the comprehensiveness of this study.

Answer

Thank you for your comment. The title was corrected. We have decided in the title not to include all the names of the pathways analysed, as this would make the title much longer. We hope that this correction will be acceptable.  

  1. The abbreviated words should be explained at their first use in the manuscript and then used consistently.

Answer

Corrections were made in line with the comment.

  1. The manuscript should be thoroughly checked for English language and grammatical errors.

Comments on the Quality of English Language: Moderate editing of English is required.

Answer

Errors in English language and grammar have been reviewed and corrected.

We are grateful for the valuable comments made by the Reviewer. The corrections made in line with them have certainly improved the manuscript.

 Sincerely,

Donatas Stakišaitis

Reviewer 2 Report

Comments and Suggestions for Authors

 Stakisaitis et al. worked on a pre-clinical study on mice to show the anti-inflammatory effects of a sodium dichloroacetate and sodium valproate combination (DCA–VPA). Mice were given DCA - VPA solution in drinking water and the anti-inflammatory effects were followed up.

There are a few questions for the authors to address before publication:

How did you follow if the mice drank a constant amount of VPA-DCA? Did you separate the mice or di you calculate the daily dose?

L. 115-116: Based on which criteria do you claim the synergistic effects of the VPA-DCA?

L. 129: Please include details about the ethical approval (e.g., no, and all further details)

Section 2.2: What is the overall duration of the study? You mentioned the duration only in the abstract and somewhere in the discussion, but in the M&M, where it should be.

L. 141: “Killing” should be replaced by “sacrificing”

L. 290-293: This part should be included in the caption of Fig. 1, otherwise commented differently.

Fig. 2 + l. 338 – 339: DCA alone and VPA alone groups are missing. How did you calculate the synergy? Please include synergy calculations.

l. 375- 3.77: Unfinished phrase.

L. 383 -401: You see a high cytokine expression even at the end of the study. This could be also an alarming effect. Too much inflammation would lead to chronic/infected wounds. Would this situation help in an infected or immunocompromised model?

Comments on the Quality of English Language

l. 375- 3.77: Unfinished phrase.

Author Response

(x) Minor editing of English language required
Answer

Errors in English language and grammar have been reviewed and corrected.

Comments and Suggestions for Authors

Stakisaitis et al. worked on a pre-clinical study on mice to show the anti-inflammatory effects of a sodium dichloroacetate and sodium valproate combination (DCA–VPA). Mice were given DCA-VPA solution in drinking water and the anti-inflammatory effects were followed up.

There are a few questions for the authors to address before publication:

How did you follow if the mice drank a constant amount of VPA-DCA? Did you separate the mice or did you calculate the daily dose?

Answer

DCA and VPA monotherapy in experimental animals has been extensively studied in various aspects and the literature indicates the concentrations of DCA or VPA monotherapy drinking solutions used in animal studies. The concentrations of DCA or VPA in the solution used in our study were chosen according to the concentrations of oral solution used by other researchers and in our previous studies. Mice were housed in a cage of 6 mice and the solution was administered to all 6 mice. We did not perform the study by keeping the mice in the metabolic cage separately and did not calculate the daily doses per mouse. Previously, we have performed studies with rats housed individually in a metabolic cage and treated with the appropriate monotherapy dose of VPA or DCA. We have published the results of these studies. We believe that the fact that we did not calculate the daily dose per mouse separately is justified. We state that we used half the concentration of DCA or VPA in this study because our studies have shown that the efficacy of DCA is increased in combination, i.e., the efficacy of half the dose of DCA in combination with VPA (e.g., anti-tumor effect) is equivalent or even more effective than treating different cells with only twice the dose of DCA. We have demonstrated synergism data in the patent for the product under investigation. The manuscript under '2.1. Investigational medicinal product' indicates this. We recorded the amount of solution drunk each day to make sure that the solution drunk was available for the mice.

  1. 115-116: Based on which criteria do you claim the synergistic effects of the VPA-DCA?

Answer

In the manuscript, we point out that the primary criterion for synergistic effects is that VPA activates the SLC5A8 carrier by epigenetic mechanisms. This effect is mediated by DNA demethylation, which leads to upregulation of the Slc5a8 gene. This is discussed in lines 78-83 and 317-329 of the revised text. We consider that the corrections made are satisfactory.

  1. 129: Please include details about the ethical approval (e.g., no, and all further details)

Answer

The data referred to in the note are given in the text of the manuscript:The use of experimental animals for the studies was approved by the State Food and Veterinary Service of Lithuania (No G2-198 of 16 March 2022)”; L: 124-125 of revised text.

Section 2.2: What is the overall duration of the study? You mentioned the duration only in the abstract and somewhere in the discussion, but in the M&M, where it should be.

Answer

The duration of treatment with DCA-VPA is mentioned subsection 2.1 of section “Material and Methods”. L114-115 of the revised text

  1. 141: “Killing” should be replaced by “sacrificing”

Answer

Adjusted in line with the comment

  1. 290-293: This part should be included in the caption of Fig. 1, otherwise commented differently.

Answer

In the subsection 3.2, the text is interchanged. We hope that the correction is consistent with the comment.

Fig. 2 + l. 338 – 339: DCA alone and VPA alone groups are missing. How did you calculate the synergy? Please include synergy calculations.

Answer

We have discussed this above. We point out that the primary criterion for synergistic effects is that VPA upregulates the SLC5A8 co-transporter by epigenetic mechanisms. This is shown by comparing the data of the treated group with the control. In the patent, we showed that DCA alone does not affect Slc5a8 gene expression, whereas VPA upregulates it. This is in our patent - studies performed on tumor cells. The fact that we do not have monotherapy groups for comparison has been added to the text on study limitations. We hope this correction is partly in line with the comment and is acceptable.

  1. 375-377: Unfinished phrase.

Answer

The sentence was corrected according to the note.

  1. 383 -401: You see a high cytokine expression even at the end of the study. This could be also an alarming effect. Too much inflammation would lead to chronic/infected wounds. Would this situation help in an infected or immunocompromised model?

Answer

Thank you for your remark. Our sequencing data show that the findings can be applied to other pathways. We see this in our studies in mice and in T lymphocytes from healthy and COVID-19 patients. This publication is the first of preclinical data. We are also preparing publications with human studies (effect of the combination on T lymphocyte gene expression).

  1. 375- 377: Unfinished phrase.

Answer. Note repeated

We are grateful for the valuable comments made by the Reviewer. The corrections made in line with them have certainly improved the manuscript.

 Sincerely,

Donatas Stakišaitis

Reviewer 3 Report

Comments and Suggestions for Authors

Overall, this pre-clinical study on the potential effects of a combination therapy of sodium dichloroacetate and sodium valproate on the management of infection pathways in mice presents promising results. The study found that the treatment did not have major impacts on the body weight or thymus weight, but did lead to an increase in the expression of the Slc5a8 gene and showed potential for modulating inflammatory mechanisms in thymocytes. One limitation of the study is that it only looked at male Balb/c mice, so the results may not be representative of other populations or genders. It would also be helpful to have a control group that received only one of the treatments to more accurately assess the individual effects of DCA and VPA. Additionally, the abstract could benefit from further clarification on the specific genes and pathways that were affected by the combination therapy. Providing more specific details on the inflammatory and immune response-related genes that were altered would make the abstract more informative. Overall, this study presents interesting findings on the potential anti-inflammatory effects of DCAVPA combination therapy and provides a strong foundation for further research in this area.

Author Response

(x) English language fine. No issues detected

Comments and Suggestions for Authors

Overall, this pre-clinical study on the potential effects of a combination therapy of sodium dichloroacetate and sodium valproate on the management of infection pathways in mice presents promising results. The study found that the treatment did not have major impacts on the body weight or thymus weight, but did lead to an increase in the expression of the Slc5a8 gene and showed potential for modulating inflammatory mechanisms in thymocytes. One limitation of the study is that it only looked at male Balb/c mice, so the results may not be representative of other populations or genders. It would also be helpful to have a control group that received only one of the treatments to more accurately assess the individual effects of DCA and VPA. Additionally, the abstract could benefit from further clarification on the specific genes and pathways that were affected by the combination therapy. Providing more specific details on the inflammatory and immune response-related genes that were altered would make the abstract more informative. Overall, this study presents interesting findings on the potential anti-inflammatory effects of DCA–VPA combination therapy and provides a strong foundation for further research in this area.

Response to Reviewer’s comments

We want to thank the Reviewer for his positive evaluation of the manuscript and for his comments. Below are the responses to the comments.

One limitation of the study is that it only looked at male Balb/c mice, so the results may not be representative of other populations or genders.

Answer

Thank you for this comment. We understand its importance. We have added this point to the section of the text where we identify the study’s limitations.

It would also be helpful to have a control group that received only one of the treatments to more accurately assess the individual effects of DCA and VPA.

Answer

In response to this comment, we have also added the limitations to the text. We understand that the comparing monotherapy versus combination effects is important for assessing synergism. We have demonstrated synergism in a patent where the combination is for the cancer treatment, and we believe that a comparing the combination with a control is sufficient in this case. Our basis for synergism in this case is the effect of VPA on the expression of the Slc5a8 carrier gene, which is the transporter of DCA into the cell. This is sufficiently demonstrated in the paper. Again, we are grateful for the comment.

Additionally, the abstract could benefit from further clarification on the specific genes and pathways that were affected by the combination therapy. Providing more specific details on the inflammatory and immune response-related genes that were altered would make the abstract more informative.

Answer

We cannot add to this note because the journal's requirements for the abstract text allow a very limited number of words.

Overall, this study presents interesting findings on the potential anti-inflammatory effects of DCA–VPA combination therapy and provides a strong foundation for further research in this area.

We are grateful for the valuable comments made by the Reviewer. The corrections made in line with them have certainly improved the manuscript.

 Sincerely,

Donatas Stakišaitis

Round 2

Reviewer 1 Report

Comments and Suggestions for Authors

It can be recommended for publication.

Reviewer 2 Report

Comments and Suggestions for Authors

Accept in present form. the authors have addressed all my comments.